# High Performance Concretes with Highly Reactive Rice Husk Ash and Silica Fume

**DOI:** 10.3390/ma16113903

**Published:** 2023-05-23

**Authors:** Andres Salas Montoya, Chul-Woo Chung, Ji-Hyun Kim

**Affiliations:** 1Civil Engineering Department, Engineering and Architecture Faculty, Universidad Nacional de Colombia, Manizales 170003, Colombia; asalasmo@unal.edu.co; 2Division of Architectural and Fire Protection Engineering, Pukyong National University, Busan 48513, Republic of Korea; 3Multidisciplinary Infra-Technology Research Laboratory, Pukyong National University, Busan 48513, Republic of Korea; kjh@pknu.ac.kr

**Keywords:** waste materials, high-performance concretes, rice husk ash, silica fume, strength characteristics

## Abstract

The search for new sources of high-quality non-crystalline silica as a construction material for high-performance concrete has attracted the interest of researchers for several decades. Numerous investigations have shown that highly reactive silica can be produced from rice husk, an agricultural waste that is abundantly available in the world. Among others, the production of rice husk ash (RHA) by chemical washing with hydrochloric acid prior to the controlled combustion process has been reported to provide higher reactivity because such a process removes alkali metal impurities from RHA and provides an amorphous structure with higher surface area. This paper presents an experimental work in which a highly reactive rice husk ash (TRHA) is prepared and evaluated as a replacement for Portland cement in high-performance concretes. The performance of RHA and TRHA was compared with that of conventional silica fume (SF). Experimental results showed that the increase in compressive strength of concrete with TRHA was clearly observed at all ages, generally higher than 20% of the strength obtained with the control concrete. The increase in flexural strength was even more significant, showing that concrete with RHA, TRHA and SF increased by 20%, 46%, and 36%, respectively. Some synergistic effect was observed when polyethylene–polypropylene fiber was used for concrete with TRHA and SF. The chloride ion penetration results also indicated that the use of TRHA had similar performance compared to that of SF. Based on the results of statistical analysis, the performance of TRHA is found to be identical to that of SF. The use of TRHA should be further promoted considering the economic and environmental impact that will be achieved by utilizing agricultural waste.

## 1. Introduction

High-performance concrete (HPC) can be defined as a type of concrete that can overcome limitations that cannot always be achieved with conventional materials and normal mixing, placing, and curing practices alone [1,2]. This can include concrete that provides increased environmental resistance (durability) and structural capacity [3]. HPC is composed of the same materials as normal concrete, but manipulated to achieve improved properties in durability, strength, or both, to meet the specific requirements of a construction project [3,4]. Therefore, the search for the benefits of HPC should focus not only on achieving improved strength and durability, but also on practical issues such as cost and sustainable construction [4]. The goal is to find the means to build structures that are economical, have better long-term performance, and have a low environmental impact [5,6,7,8].

HPC could be produced from Portland cement alone or from a combination of cement and mineral admixtures, which are used to improve the properties of concrete in both the fresh and hardened states [9]. By replacing, even partially, Portland cement with mineral admixtures, both economic and environmental benefits are achieved by reducing and lowering the amount of CO_2_ emissions from the cement industry [10,11,12,13,14,15,16], one of the world’s most polluting industries [2,4,7,17]. The mineral admixtures, depending on their nature, impart different properties to the concretes, which are generally positive, especially from the point of view of the durability of the mortars and concretes [18,19,20,21,22].

Pozzolanic materials, for example, have a double effect when combined with Portland cement: they exert a chemical reaction in the cement paste and a filling action that causes the closure of pores and voids [19,23,24,25]. According to Wang and Jia [18], the filling effect is of greater importance at the microstructural level of the concrete, not only because it has the function of closing the pores, but also because the small grains act as nucleation points for the precipitation of hydration products, allowing greater densification and strength [26]. The physical action (filling) of the pozzolans provides a more uniform, dense, and homogeneous paste, which has contributed to obtaining concretes with properties far superior to those obtained in normal Portland cement concretes. Among the admixtures for high-performance concrete, fly ash, slag, and silica fume have been the most widely used alternative cementitious materials [3].

Silica fume (also called micro-silica; SF) is one of the most active pozzolanic materials and one of the most widely used admixtures in high-performance concretes [27,28,29]. When used in combination with high-range water-reducing admixtures, it has allowed a significant increase in the strength and durability of concretes, not only because of its extremely rapid pozzolanic reaction but also because of its high fineness, which provides a beneficial contribution to the concrete, particularly by densifying the mix and reducing permeability levels, thus contributing to both its mechanical behavior and durability [2,27,30]. Despite being an industrial by-product, SF can be several times more expensive than Portland cement and its availability is limited worldwide [31].

Rice husk ash (RHA), obtained from the controlled burning and grinding of rice husks, a by-product of an agricultural process, is classified as another highly reactive pozzolanic material [32] that can be used as an alternative cementitious material, partially replacing Portland cement in concrete mixes [33,34]. This mixture promotes a pozzolanic reaction with calcium hydroxide, forming calcium silicate hydrate (C-S-H), reducing voids, and promoting a reduction in permeability, which positively affects both the mechanical strength and durability of the concretes [32,34,35]. The most important properties governing the pozzolanic activity of RHA are the amorphous phase content and the specific surface area, both of which are closely related to the firing conditions of the husk [34]. This is a very sensitive process, especially because burning rice husks at temperatures below 600 °C produces silica but with residues of unburned carbon, and at temperatures above 700 °C silica crystallizes [36,37]. In both cases, the reactivity of the ash is severely compromised [19,38,39]. This sensitivity to combustion conditions has not allowed a massive use of this additive. Some authors [38,40,41] have found that by acid washing the rice husk before firing, it is possible to obtain relatively pure silica with a specific surface area even greater than that of silica fume and with superior pozzolanic properties, which also reduces the sensitivity of this material to firing conditions [42,43,44].

Acid washing of rice hulls removes a high percentage of impurities that contribute to the crystallization of silica [44]. According to Vayghan et al. [40], this treatment increases the specific surface area and reactivity of the ash without affecting the amorphous nature of the silica. However, the effect of this type of ash on the properties of high-performance concrete has not been studied in detail. There are only a few studies on RHA with acid washing [38,42,44,45,46,47].

It should be noted that RHA is an organic waste source with reactive silica that is abundantly available worldwide and could be used as a cementitious admixture with superior pozzolanic properties than silica fume (SF) for the development and application of HPC. Well-manufactured RHA can be an ideal alternative to SF, as SF can be several times more expensive than RHA due to the critical shortage of SF. However, the importance of RHA as a reactive pozzolanic material has not been emphasized as much as SF and metakaolin, and therefore there are much fewer studies reported in the literature. A direct evaluation of the performance of cementitious composites between RHA with and without acid washing, as well as a comparison of the performance of RHA with SF, has not been well documented.

To address this need, this paper presents the evaluation of the effect of two RHA, one thermally treated and the other chemically and thermally treated, on the mechanical and durability properties of high-performance concrete mixes, comparing the results with those obtained in mixes containing SF at the same replacement levels. In the experimental phase, high-performance concrete mixes were designed and prepared to evaluate their compressive strength, modulus of elasticity, flexural strength, flexural toughness, and durability properties. Some cement pastes were also prepared to determine the hydration products and understand their effect on the final properties of the concrete.

## 2. Materials and Methods

### 2.1. Cementitious Materials

A commercial type V Portland cement (PC), compatible with ASTM type I Portland cement [48], without limestone powder, was used for all concrete mixes. In addition, two different types of RHA’s were used as a partial replacement of PC. One of the ashes (RHA) was prepared by burning a rice husk from a local rice mill at a temperature of six hundred degrees Celsius for three hours; the second ash (TRHA) was prepared by immersing a sample of rice husk in a hydrochloric acid solution at a concentration of 1 normal for twenty-four hours, followed by thermal treatment in a furnace at six hundred degrees Celsius for three hours [41]. Both ashes produced were ground in a ball mill to increase fineness and reactivity. To compare the results of the concrete mixes with the two RHAs, silica fume (SF) was also used as a PC replacement in the same percentages. Table 1 and Table 2 show the chemical composition and physical properties of the supplementary cementitious materials used in this research, respectively.

Previous tests [49] showed that TRHA is an almost 99% amorphous pure silica with a specific surface area of 274,000 m^2^/Kg, whereas RHA had 24,000 m^2^/Kg. This extremely high specific surface area could be due to an internal pore network originally derived from the rice husk. Other researchers [38] have found that acid treatment causes a significant increase in specific surface area, creating a mesoporous internal structure. When rice husk is treated, the K+ ions present in the husk, which are responsible for the surface melting of the ash particles, are leached out, allowing the formation of mesopores. Figure 1 shows the shape and size of some RHA particles.

### 2.2. Aggregates

Table 3 shows the gradation of the aggregates used in this study. Natural sand was used as fine aggregate (FA) with a fineness modulus of 3.05. The coarse aggregate (CA) was calcareous limestone with a maximum grain size of 25.4 mm. The gravel and the sand were mixed to adjust the gradation to the limits recommended by ASTM C33 [50] to produce workable and compact concrete mixes.

### 2.3. Superplasticizer

Due to the high specific surface area of the RHA and SF, it was necessary to use a superplasticizer. In this study, a polycarboxylate-based superplasticizer was used. Municipal drinking water was used for all concrete mixes.

### 2.4. Fibers

Polymeric fibers were added to the fiber-reinforced concretes to evaluate their mechanical properties. For this purpose, commercial polymeric fibers were used, whose commercial brand is STRUX 90/40, and whose specifications, provided by the manufacturer [51], are shown in Table 4. The fiber dosage used for all the concrete mixes was 2.38 kg/m^3^.

### 2.5. Mixture Proportions

#### 2.5.1. Cementitious Pastes

To evaluate the effect of mineral additions on Portland cement hydration processes, cementitious pastes were prepared with the two RHAs and SF. The mineralogical composition and microstructure of the products formed during these hydration processes were evaluated using X-ray diffraction (XRD) and scanning electron microscopy (SEM) techniques. The cementitious pastes contained percentages of 0% (control) and 10% of each of the mineral additions as partial replacements for Portland cement. All pastes were prepared with a water/cementitious materials ratio of 0.45 and the mixing and curing procedures were carried out according to ASTM C 305 [52]. The products generated by the hydration reactions between the Portland cement and RHA or SF were identified and characterized after 90 days. The porosity of the cementitious pastes was determined by mercury intrusion porosimetry (Micromeritics Autopore II 9220), with an automated system that measures pore size distribution with diameters in the range between 300 µm and 3 nm.

#### 2.5.2. Concrete Mixtures

Concrete mixes containing RHA, TRHA, and SF were designed and produced following a previous study [45,49]. RHA, TRHA, and SF were replaced by 10%, and an average workability of 100 to 150 mm was maintained with a superplasticizer, as determined by the Abrahms cone test according to ASTM C143 [53]. For comparison purposes, the water/cementitious material ratio was kept constant at 0.45 for all mixes, as well as the amount of cementitious material, which was 440 kg/m^3^ and a water content of 198 L per cubic meter. Two sets of concrete mixes were evaluated, one set of plain concrete mixes and one set of mixes reinforced with polymer fibers. All concrete mixes were produced using the same parameters as specified in Section 2.5. The aggregate content was adjusted according to the design type specified in ACI 211.1 [54]. In total, eight mixes were prepared. The proportions of each mix are shown in Table 5 below.

### 2.6. Mixing and Casting Procedures

The concrete was mixed in a laboratory tilting drum mixer for a total of nine minutes. The mixing procedure was in accordance with ASTM C192 [45]. Immediately after mixing, a slump test was performed to measure the workability of each mix. Cylinders measuring 100 × 200 mm were cast from each mix for compressive strength (f’c), modulus of elasticity, and resistance to chloride ion penetration, and beams measuring 150 × 150 × 500 mm were cast for flexural strength (ft). After casting, the concrete specimens were covered and stored under controlled temperature and humidity for 24 h. The specimens were then demolded, marked, and placed in a curing room until the time of testing.

### 2.7. Testing of Concrete Samples

The mechanical performance of the hardened concrete was tested according to ASTM standards. Compressive strength was determined using 100 × 200 mm cylinders according to ASTM C39 [55] at 28, 56, 90, and 180 days of curing, flexural strength expressed as modulus of rupture (MR) was determined by the standard ASTM C 78 [56] (third point loading) test method using four beams per mix at 56 days of curing, and Young’s modulus at 56 days of curing was determined according to ASTM C469 [57] using five cylinders per test day. Fracture toughness analysis of both plain and fiber concrete beams was performed according to the method described in ASTM C 1018 [58]. Resistance to chloride ion permeation was measured according to ASTM C1202 [59] after 56 days of curing; these tests were performed on cylindrical specimens 100 mm in diameter and 50 mm in length. For the carbonation test, specimens with a diameter of 75 mm and a height of 150 mm were subjected to accelerated carbonation.

## 3. Results

### 3.1. Effect of SCM’s on the Hydration of Cement Pastes

Figure 2 and Figure 3 show the X-ray diffractograms and electron microscope micrographs obtained for each of the cementitious pastes after 56 days of curing; the main hydration products in the cementitious pastes, indicating the Bragg angle values (2θ) of their characteristic peaks from the XRD analysis, have been identified in Figure 2. As can be seen, there is a very marked difference between the intensities of the portlandite peaks at 56 days for the different blended mixes.

It can be observed in Figure 2 that the RHA-added pastes have comparatively lower peaks in the portlandite intensities for the standard mix, but they are higher than those of the silica fume and the treated husk ash, which is caused by the lower pozzolanic reactivity of the untreated ash, as it can be seen on the results of the pozzolanic activity index shown in Table 2. Figure 3 shows SEM images of the control paste and the pastes with 10% replacement, but they showed similar characteristics. In the control paste, the presence of large CH crystals embedded in a dispersed matrix of CSH gel can be observed; in the pastes with pozzolanic replacement, the presence of more compact CSH gel and almost imperceptible CH crystals can be observed.

#### Porosity of the Cementitious Pastes

Table 6 and Figure 4 and Figure 5 present the results of the comparative study on the pore structure of the different cementitious pastes, showing the evolution of the porosity at 56 and 90 days of age; the results have been expressed as a percentage of the total porosity.

Figure 4 and Figure 5 show that after 56 days there is a refinement of the porous structure of the pastes with the mineral admixtures when compared to the pore structure of the control paste, represented by a significant decrease in the content of capillary pores (those with sizes greater than 0.1 microns) as mentioned previously [26,44]. Such reduction becomes more significant in SF and TRHA pastes and is accompanied by an increase in the number of intermediate-size pores (those with sizes between 0.1 and 0.01 microns, located between the gel pores and the capillary pores) and a smaller increase for the pore content between 0.01 and 0.0025 microns.

A statistical analysis of the results was performed with Tukey’s post-ANOVA test to compare the means of the mixtures of the pore diameter variable. It was found, with a 5% significance level (95% confidence level), that the TRHA and SF pastes presented a smaller average pore diameter and a higher porosity compared to the data of the control paste; these results are similar to those presented by Poon et al. [60], who used silica fume as a substitute for Portland cement in cementitious pastes with a w/cm ratio of 0.45. Similarly, Hwang et al. [61] mentioned the decrease in average pore size that occurs in pastes with RHA, at percentages of 5 and 15% as a PC replacement. This refinement of the porous network of the pastes can be attributed to the segmentation of the larger capillary pores caused by the precipitation of the hydration products generated by the pozzolanic reactions between the silica of the admixtures and the calcium hydroxide released by the hydration of the Portland cement. As the spaces between the hydration products are filled, the larger pores become smaller, subdividing, and increasing the content of intermediate-size pores [62].

### 3.2. Effect of SCMs on Simple and Fiber-Reinforced Concrete Mixes

#### 3.2.1. Workability

Figure 6 shows the slump test results and superplasticizer requirements of the unreinforced and fiber-reinforced concrete mixes. The control mix (100% Portland cement) did not require any admixture dosage to achieve the highest slump of all the mixes studied, whereas the treated RHA mixes required a higher superplasticizer dosage to achieve relatively low slumps for a very short duration as observed by Eldale et al. [36], which made the filling process of some of the molds very difficult.

Similarly, the high superplasticizer requirements of the RHA mixes, although lower than those of the treated ash mixes, are higher than those of the silica fume mixes. During the preparation of the mixes with RHA and SF, there were no problems of rapid loss of slump, as occurred in the mixes with TRHA. All the fiber-reinforced mixes, with and without admixtures, had higher water requirements than the plain concrete mixes. According to previous studies [63,64], the addition of any type of fiber reduces the consistency of concrete mixes, especially in conventionally mixed fiber-reinforced concrete, making it necessary to use higher doses of superplasticizer to achieve sufficient mix consistency without increasing the water to cementitious materials ratio.

#### 3.2.2. Compressive Strength

The results of the compressive strength tests are shown graphically in Figure 7. Results are also summarized in Table 7. The highest compressive strength values of all unreinforced and reinforced concrete mixes are obtained with the partial replacement of PC by TRHA. Similarly, the mixes containing RHA have higher compressive strength than the control mix at all ages.

The maximum value of the compressive strength of the unreinforced concrete was 58 MPa for the mix with TRHA at 180 days, whereas the maximum strength for the SF mix was 57 MPa at the same curing age. These results with SF are comparable to those reported previously [49,63,64,65,66], who produced concrete mixes under conditions and parameters similar to those used in this work, but in the case of fiber-reinforced concrete incorporating untreated and treated husk ash, there are no references to date with which to compare the results obtained here.

Although the increase in compressive strength of the mixes with TRHA is not statistically significant compared to the SF mixes, it can be said that this chemically treated ash could be a substitute for SF as a supplementary cementitious material to obtain concretes with better performance than normal mixes. These results show the high reactivity of the TRHA, especially in the early stages of hardening, which can be observed in Figure 2, comparing the intensities of the peaks corresponding to portlandite (CH) in the cement pastes, where the lowest intensity corresponds to the peaks of the pastes with TRHA, indicating a higher consumption of calcium hydroxide than in the pastes with SF and RHA. Comparing the compressive strength values of the unreinforced and fiber-reinforced concrete mixes, no significant effect of the incorporation of polypropylene fibers on compressive strength is found. There is only a slight reduction in strength which can be attributed to the higher porosity of the fiber-reinforced mixes. These data are corroborated by some bibliographical references [67,68], which state that the compressive strength is relatively little affected by the addition of this type of fiber to concrete mixtures.

Statistical analysis of the data obtained was performed using the Tukey test to compare the means of the mixtures. It was found that the means within each group were statistically similar at the 5% level of significance. It can be observed that the samples with TRHA were the group with the highest compressive strengths. It was also observed that the fiber-reinforced TRHA sample was statistically equal to the mixture with SF and that the mixtures with RHA were statistically equal to the control mixtures. This gives an indication of how these additions can act as a replacement for both SF and Portland cement, respectively.

#### 3.2.3. Elastic Modulus

The modulus of elasticity of the concrete mixes was calculated according to ASTM C469 [57]. The test was performed at a curing age of 56 days, applying a load equal to 40% of the maximum strength of each cylinder. This test was performed at a loading rate that was maintained within the range of 0–2–0.24 MPa/sec. Figure 8 shows the results for the plain and the fiber reinforces mixes. For all the concretes containing the RHA and SF, the Young’s modulus was higher than the control mix. As in the compressive strength test, the treated husk ash (TRHA) concrete mix had the highest Young’s modulus of all the mixes, 32.1 GPa, compared to 31.8 GPa for the SF mix, 30.2 GPa for the untreated ash, and 29.1 GPa for the control mix. As for the results obtained for the fiber-reinforced mixes, the mixes with SF and with TRHA showed very similar behavior; the test yielded a value of 30.8 and 30.9 GPa, respectively, for each sample, compared to 29.4 MPa for the mix with RHA and 28.0 MPa for the control mix.

Figure 8 also shows how the incorporation of fibers into the concrete mixes affects the modulus of elasticity of the concrete mixes. The lower values obtained by the fiber-reinforced mixes are related to, among other things, the higher porosity of the fiber-reinforced concrete and the lower modulus of elasticity of the fibers. According to existing literature [69,70], the addition of polypropylene fibers to concrete mixes increases the pore content of this material and, therefore, there is a loss of mechanical strength. This would explain the reduction in the elastic modulus of the material, which is very sensitive to the presence of microcracks and pores, as well as the reduction in compressive strength [67]. In the bibliographic references consulted, no information was available on the modulus of elasticity for fiber-reinforced concrete mixes with RHA or for plain and fiber-reinforced concrete mixes with TRHA.

The statistical analysis of the variable modulus of elasticity, the results of Tukey’s multiple comparison tests with a significance level of 0.05, showed that the mixes with TRHA produced the highest strength performance among all the mixes, especially when compared to the control mix. It was also found that the results obtained between the mixes with TRHA and SF are comparable, which means that the addition of SF can be replaced by the addition of TRHA. The statistical analysis also shows that the control and RHA blends are statistically similar, so it is possible to state that the replacement of this addition by PC does not have any negative consequences, on the contrary, the effect on the modulus of elasticity is positive, representing an increase in its magnitude. As for the effect of the addition of fibers on the results of the modulus of elasticity, it can be observed that they do not play a significant role in the values of this property, since the increase in the modulus of elasticity does not make the samples statistically different from the plain concrete mixes.

#### 3.2.4. Flexural Strength

Figure 9 shows the calculated values of the flexural strength (modulus of rupture) for the concrete mixtures, following the procedure established in ASTM C 78 [56]. The effect of the incorporation of pozzolans on the flexural strength of the concrete mixtures is similar to the effect that occurred with the compressive strength. The incorporation of RHA increased the flexural strength by 20% over the control mix, whereas SF and TRHA increased the strength by 36 and 46%, respectively. Regarding the effect of fibers, increases in strength of 11% for the control mixes, 7% for the mix with untreated husk ash, 4% for the mix with silica fume, and 3% for the mix with treated husk ash were observed. These data indicate the modest influence that the incorporation of polypropylene fibers has on these types of mixtures. These results are in correspondence with other investigations carried out on mixtures with and without reinforcement [71,72,73], which reported how flexural strength increases while compressive strength increases. This increase in the added mixtures is related to a refinement in their porous structure, in a less dense and therefore stronger interfacial transition zone, as mentioned by Justice [74].

Comparing the flexural strength results of the control mixes with those of the fiber reinforced concretes, the combined effect of the pozzolanic admixtures and polypropylene fibers on flexural strength becomes significant. The increases in flexural strength were 28% for the RHA mix, 41% for the SF mix, and 50% for the TRHA mix, so it can be said that there was a good interaction between the fibers and the admixtures.

Tukey’s statistical test for multiple comparisons shows, at a 5% level of significance, that the highest performance in flexural strength is presented by the fiber-reinforced mixes with TRHA and SF, respectively, and in third place by the unreinforced concrete mix added with TRHA. As in the modulus of elasticity test, the control and RHA samples are statistically similar, as well as the samples with TRHA and SF. Figure 10a shows the stress-deflection curves for the unreinforced mixes, and Figure 10b shows the corresponding curve for the fiber-reinforced mixes.

Figure 10a,b show the behavior of the unreinforced mixes only up to the value of the maximum load, since after this point there is an abrupt drop in the stress-deflection graph, whereas, with the presence of fibers in the mixes, there is a greater load capacity in the specimens, which leads to a better performance of the toughness of the mixes beyond the value corresponding to the maximum resistance. As discussed above, the maximum stresses carried by the fiber-reinforced beams were slightly higher than those carried by the plain concrete beams, showing an increase in the value of the modulus of rupture. The analysis of Figure 10 also shows the superior performance of the mix with RHA, both for the unreinforced and the reinforced mixes, although the latter showed a more brittle behavior than that of the mixes with SF, with a more pronounced drop in the post-maximum load portion of the curve. Meanwhile, the mix with RHA showed an intermediate behavior between the control mix and the SF mix.

#### 3.2.5. Flexural Toughness

As mentioned above, the fracture toughness analysis of both plain and fiber-reinforced concrete beams was performed according to the methodology described in ASTM C 1018 [58]. This standard uses dimensionless parameters to describe the toughness performance of concrete mixes. These parameters, called “toughness indices”, are calculated as the ratio of the energy absorbed by the specimen at a given deflection (a multiple of the deflection at the first crack) to the energy absorbed by the material until the first crack appears. The numerical calculation of these toughness indices is performed from the stress-deflection curve and is expressed as the area under this curve up to a given point, which can be 3, 5.5, 10.5, 15.5, and 30.5 times the deflection at which the first crack occurs for the most used toughness indices, I5, I10, I20, I30, and I60.

The values of the toughness indices are indicative of the elastic behavior of the material; thus, results of 5, 10, 20, 30, and 60 for the toughness indices I5, I10, I20, I30, and I60 represent a linear elastic evolution of the material before the appearance of the first crack and a pseudoplastic development after this appearance. Similarly, values close to 2 for the ratios I10/I5, I20/I10, I60/I30, and 1.5 for the ratio I30/I20 also indicate a perfectly plastic behavior of the material between the deflections associated with these ratios; values lower than these indicate a lower behavior of the material. In the case of unreinforced concretes, the results of all the toughness indices are equal to 1, due to the total absence of the load-bearing capacity of the material.

Table 8 and Figure 11 show the toughness indices obtained from the stress–strain curves of each of the concrete mixes analyzed. The test results show that the mix with TRHA exhibited the most unfavorable behavior in terms of energy absorption capacity. As can be seen in Figure 10a, the curve corresponding to this mix shows an abrupt drop in the area after the first crack (according to ASTM C1018, where the slope of the curve shows a definitive change), which corresponds to the maximum peak of the curve. This performance is related to the loss of load-bearing capacity of the material due to its brittleness and the poor adhesion between the fibers and the matrix [75].

The values corresponding to the toughness indices of the concrete beams with SF were higher than those of all the mixtures with pozzolanic materials, although they were close to those obtained for the concrete with RHA, and the highest values were found in the control mixtures without admixtures. The relationships between the toughness indices do not indicate a response close to an elastoplastic behavior of the added materials, partly due to the low effectiveness of the reinforcing fibers, which did not allow large deflections of the material due to their smooth texture, their low modulus of elasticity and their low percentage content.

#### 3.2.6. Durability Tests

The unreinforced and reinforced concrete mixes were tested for the following durability properties after 56 days of curing: resistance to chloride penetration, ASTM C1202 [54], and concrete carbonation tests, in which the concrete specimens were subjected to accelerated carbonation.

##### Chloride Ion Permeability

The results of the chloride penetration resistance test are shown in Table 9. The results obtained in the test showed that the blended mixtures greatly outperformed the control mixtures. It is observed that the control sample developed a chloride ion permeability of 3529 coulombs at 56 days of age, which is considered moderate permeability according to ASTM C 1202 [54]. Meanwhile, the concrete with TRHA developed a very low chloride ion permeability (960 coulombs) and the silica fume mix developed a very similar value (970 coulombs). This is mainly since, as mentioned above, the incorporation of these pozzolanic admixtures into the concrete results in a much finer porous structure in the hydrated paste, especially at the aggregate/paste interface.

##### Carbonation Tests

Samples with a diameter of 75 mm and a height of 150 mm were subjected to accelerated carbonation; the samples were placed in a storage room at 29 °C, 3% CO_2_ content, and %HR: 65–70%. After 56 days, the specimens were cut in half and the carbonation depth was determined by spraying a fine mist of an aqueous solution of 1% phenolphthalein in 70% ethyl alcohol, used as a pH indicator, on the broken surface of the specimens [76]. The carbonation depths for the different mixtures are presented in Table 10. The differences in the results between the mixtures are not significant with respect to the control mixture, results similar to the obtained by Sanjuán et al. [76], who experimented with mixtures containing fly ash and silica fume under conditions of exposure to severe and natural carbonation and with different water/cementitious materials ratios.

## 4. Discussion

In this work, experimental work was planned to investigate the effect of a highly reactive RHA and SF on the mechanical and durability performance of high-performance concretes. According to our work, although the addition of SF, RHA, and TRHA led to a significant reduction in the workability of fresh mixes, the incorporation of these pozzolanic materials increased the compressive and flexural strength as well as the modulus of elasticity of concrete. For compressive strength, the TRHA showed the highest strength at all curing ages. At 28 days, the strength was about 17% higher than the control mix, 12.3% higher than concrete with the RHA, and 3% higher than concrete with the SF; at 180 days. No statistical differences were found when comparing the results between the TRHA and SF mixes, and the increase was 10% and 12% with respect to the RHA and control mixes, respectively. In general, the performance of TRHA was identical to that of SF (in terms of experimental data with statistical analyses), but the performance of RHA was not as superior to that of TRHA.

The effect of the incorporation of SF and TRHA on the flexural strength of concrete mixes is similar to the effect on the compressive strength. In plain concretes the incorporation of RHA increased flexural strength by 20% over the control mix, whereas SF and TRHA increased strength by 36 and 46%, respectively. The combined effect of the pozzolanic admixtures and the polypropylene fibers on the flexural strength of the control concrete is of the order of 28% for the mix with RHA, 41% for the mix with SF, and 50% for the mix with TRHA. All these experimental results indicate that there is a good synergetic interaction between the polymeric fibers and SF, RHA, and TRHA.

It should be noted that incorporation of the RHA into the pastes and concretes would reduce the porosity and Ca(OH)_2_ content in concrete by the pozzolanic reaction. However, unlike the case of RHA, TRHA has a higher surface area than RHA with less crystallization due to the leaching of K^+^ from the rice husk structure during the acid washing process. Such a difference caused TRHA to provide more sites for Ca(OH)_2_ to react with, resulting in a higher efficiency to refine pore structure, reduce porosity, reduce Ca(OH)_2_ content, and increase the amount of C-S-H in the concrete matrix. Due to such beneficial effects, concrete with TRHA showed the lowest chloride ion penetration, which is identical to that with SF.

It is concluded that the performance of TRHA as a pozzolanic material is identical to that of SF. As noted earlier, the price of SF is one of the most expensive materials to produce concrete, so the use of RHA should be more facilitated considering the economic benefits as well as the standpoint of agricultural waste disposal. Burning the husk is not an environmental issue because rice absorbs CO_2_ during growth (becomes carbon neutral). The authors strongly believe that the use of TRHA can be an ideal alternative to SF when the method to reduce the amount of acid waste is developed.

## 5. Conclusions

According to the experimental results obtained from this work, the following conclusions can be drawn:
THRA is a highly reactive pozzolanic material that, when added to concrete mixes, produces superior results in both strength and durability.The pozzolanic reactions that occur with the incorporation of RHA in cementitious mixtures generate denser hydration products (CSH gel) that eliminate most of the capillary porosity, causing the formation of finer pores, resulting in a refinement of the porous network of the material, reducing its permeability and increasing its strength and durability properties.The increase in compressive strength of the mixes with TRHA is more pronounced at short ages, with values higher than 20% of the strength obtained with the control mix. This high initial reactivity is due to the high specific surface area and the higher purity of silica in the ash. At higher ages, the increase in strength is comparable to that caused by SF, with percentage values 12% higher than the strength of the control mix.The incorporation of RHA increased flexural strength even more significantly. The flexural strength of concrete with RHA, TRHA, and SF increased by 20%, 46%, and 36%, respectively.The incorporation of polyethylene–polypropylene fibers into the HPC mixtures does not significantly affect the values of compressive strength and modulus of elasticity of the concrete mixtures. However, for the flexural strength, it increases up to 50% for the mixture with TRHA compared to the strength of plain concrete and is of the order of 28% for the mixture with RHA and 41% for the mixture with SF.The results of the chloride ion penetration and carbonation test were negatively affected by the addition of polyethylene–polypropylene fibers due to the higher porosity in the interface between polymeric fiber and cement paste.Despite TRHA is a material with pozzolanic properties that is identical to SF, it is necessary to perform further research to demonstrate its effectiveness under different environmental conditions. Such activity is important because this agricultural waste is abundantly available in several countries of the planet.

## Figures and Tables

**Figure 1 materials-16-03903-f001:**
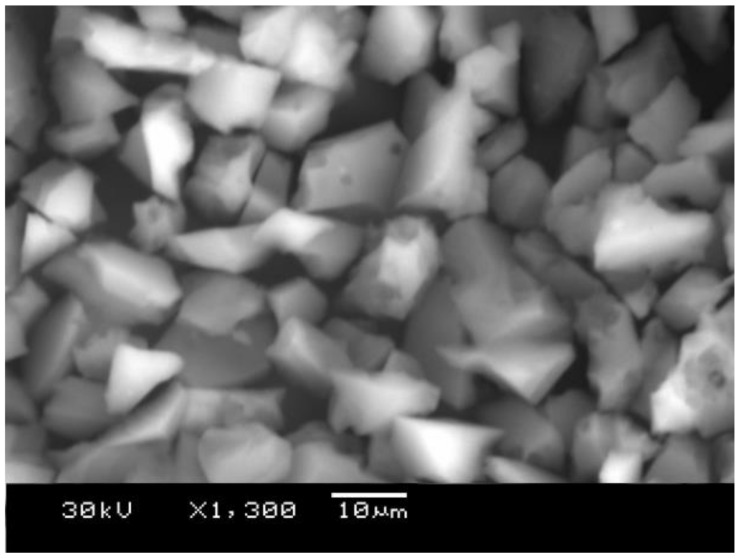
Treated rice husk ash particles.

**Figure 2 materials-16-03903-f002:**
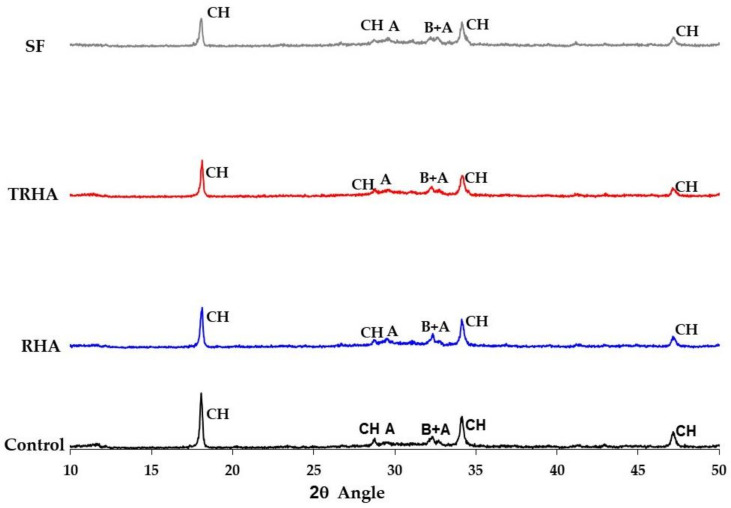
X-ray diffraction patterns for cementitious pastes at 56 days. Characteristic peaks of chemical components of cementitious pastes (XRD): CH: calcium hydroxide (Portlandite). A: tricalcium silicate (Alite); B: dicalcium silicate (Belite).

**Figure 3 materials-16-03903-f003:**
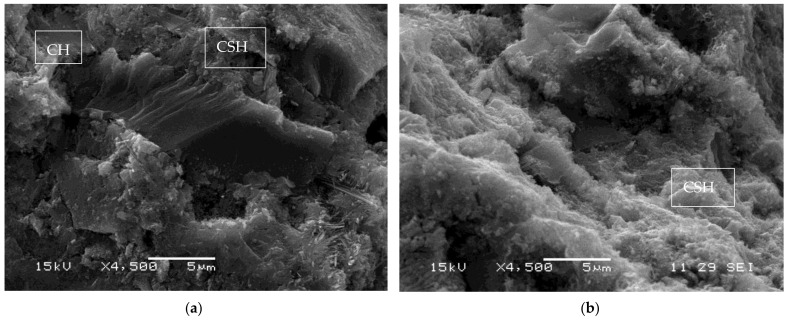
SEM images of pastes at 90 days of age: (**a**) Control paste (**b**) 10% SF (**c**) 10% RHA (**d**) 10% TRHA.

**Figure 4 materials-16-03903-f004:**
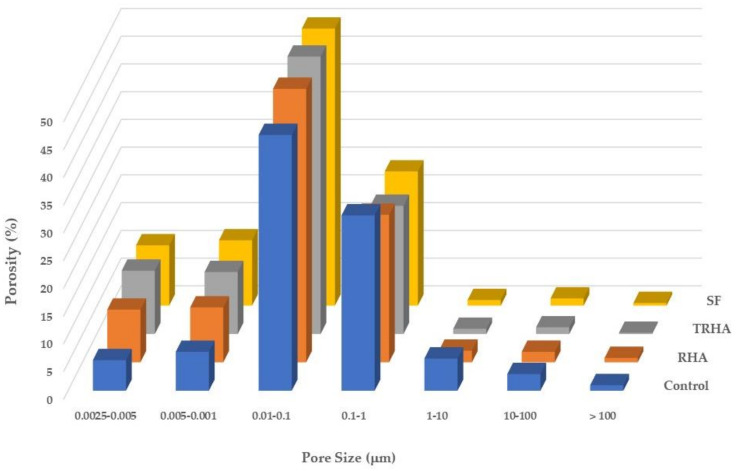
Porosity of cementitious pastes at 56 days of age.

**Figure 5 materials-16-03903-f005:**
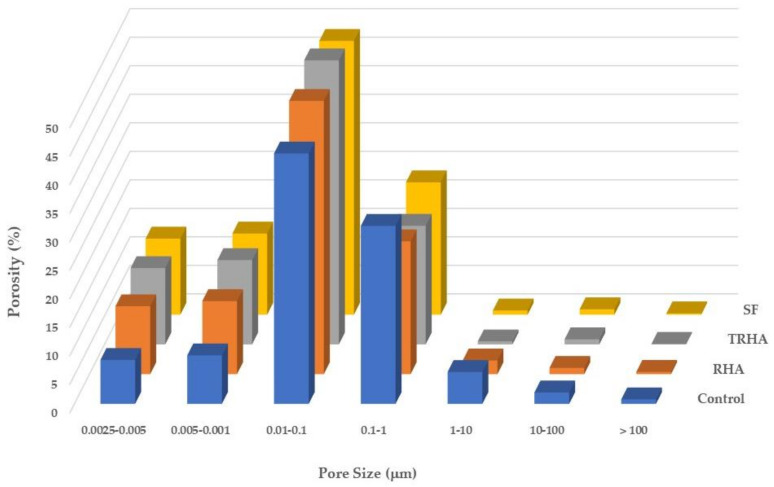
Porosity of cementitious pastes at 90 days of age.

**Figure 6 materials-16-03903-f006:**
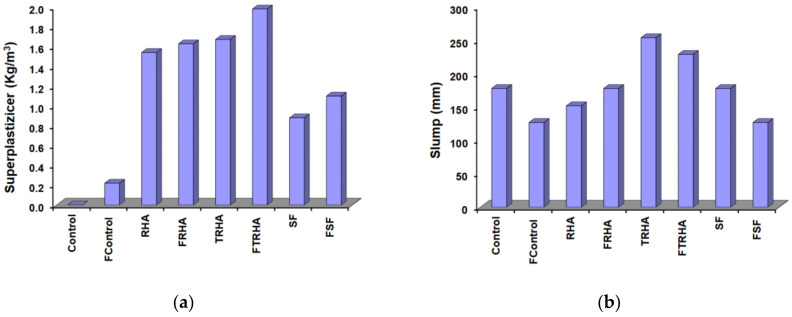
(**a**) Plasticizer required to maintain constant workability and (**b**) Variation in slump of concrete mixes.

**Figure 7 materials-16-03903-f007:**
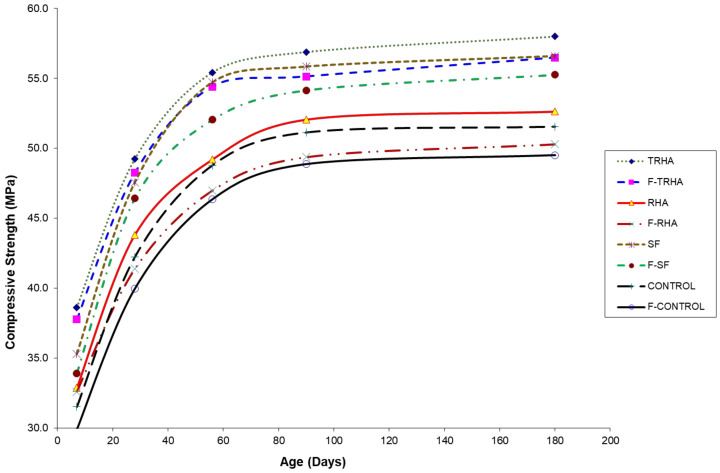
Compressive strength of plain and fiber-reinforced concrete mixtures at different curing ages.

**Figure 8 materials-16-03903-f008:**
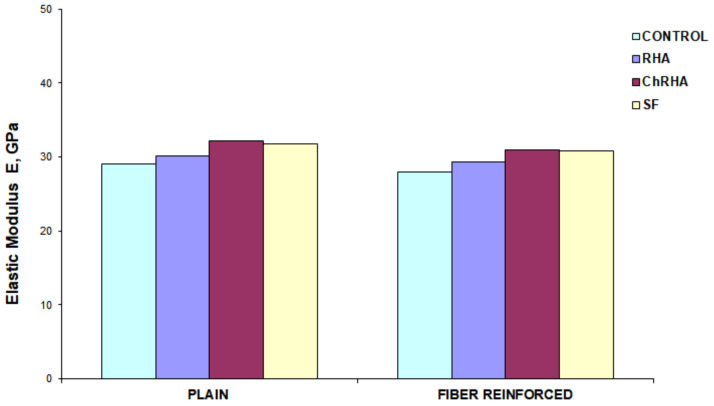
Modulus of elasticity of plain and fiber-reinforced concrete mixtures as a function of incorporation of different pozzolanic admixtures.

**Figure 9 materials-16-03903-f009:**
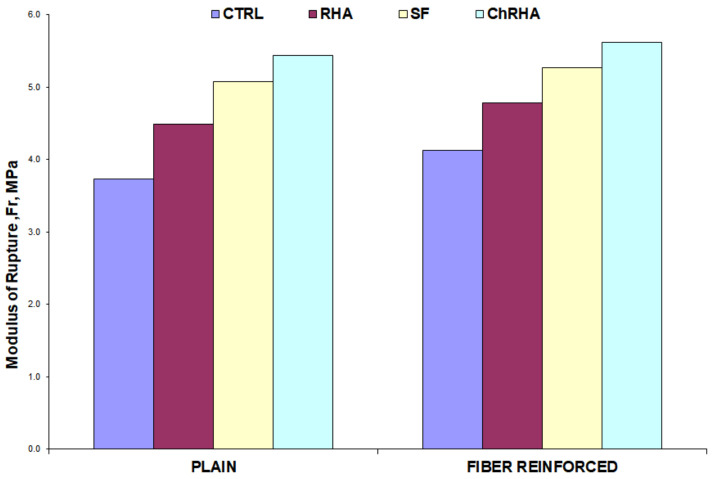
Flexural strength of plain and fiber-reinforced concrete mixtures.

**Figure 10 materials-16-03903-f010:**
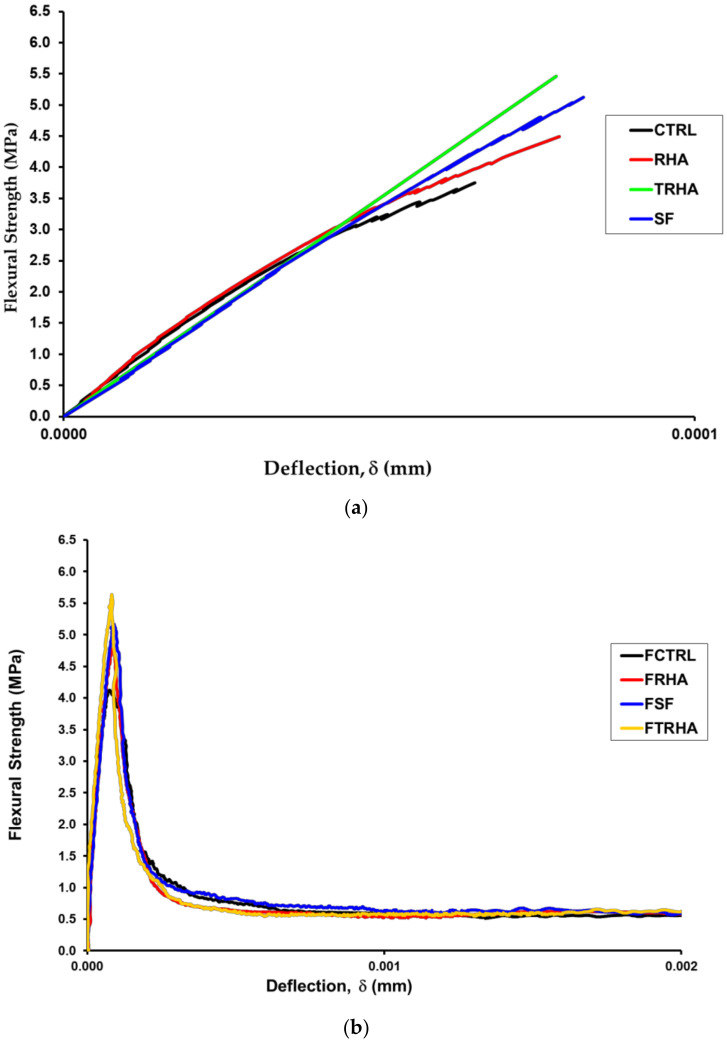
Stress-deflection curves for (**a**) simple and (**b**) fiber-reinforced concretes with and without pozzolanic admixtures.

**Figure 11 materials-16-03903-f011:**
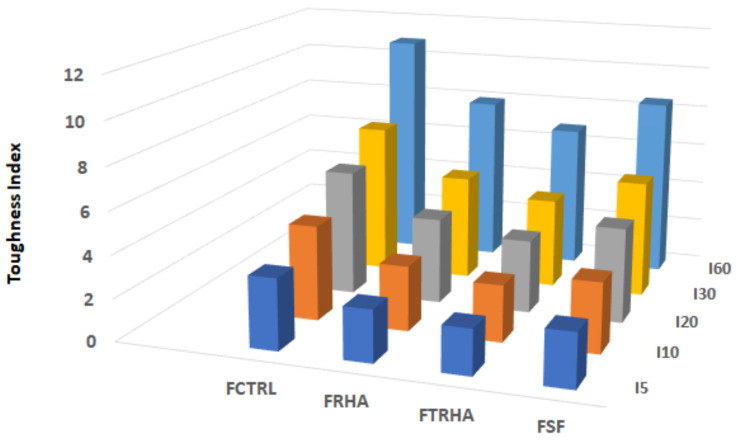
Toughness indices for fiber-reinforced mixtures with and without pozzolanic additions.

**Table 1 materials-16-03903-t001:** Chemical compositions of cementitious materials used.

	SF	TRHA	RHA	PC
SiO_2_	90	99	90	21.27
Al_2_O_3_	0.46	<0.01	0.68	4.63
Fe_2_O_3_	4.57	0.13	0.42	3.96
K_2_O	1.69	0.06	2.8	0.18
CaO	0.49	0.49	1.23	63.05
MgO	0.68	<0.07	0.35	1.56
Na_2_O	0.05	<0.32	<0.32	0.16
LOI (%)	0.54	0.16	0.13	2.25

**Table 2 materials-16-03903-t002:** Physical characteristics of cementitious materials used.

	SF	TRHA	RHA	PC
Specific gravity	2230	2080	2160	3050
Specific surface, (m^2^/Kg)	27,000 (BET)	274,000 (BET)	24,000 (BET)	377 (BLAINE)
Median grain size, (μm)	0.1	17	19	15
Color	Dark Gray	White	Pink	Gray
Pozzolanic Activity Index ASTM C618-ASTM C311	123	125	95	-

**Table 3 materials-16-03903-t003:** Gradation and physical properties of aggregates.

	Passing Percentage (%)		
Sieve Size (mm)	Coarse Aggregate	Coarse Sand	Fine Sand
19.05	100	100	100
12.7	63.6	100	100
9.51	31	96.6	100
4.76	8.4	81.3	99.9
2.38	5.7	66.2	97.5
1.19	4.1	49.5	90.6
0.595	3.7	27.9	79.8
0.297	3.3	9.5	48.7
Fineness modulus		3.63	1.78
Volumetric density (g/cm^3^)	2.98	2.75	2.71
Abs. capacity %	1.61	1.64	2.7

**Table 4 materials-16-03903-t004:** Properties of STRUX Fiber 90/40.

Specific Gravity (kg/m^3^)	920
Absorption, % (kg/m^3^)	None
Modulus of Elasticity	9.5 Gpa
Tensile Strength	620 Mpa
Melting Point	160 °C
Flash Point	590 °C
Chemical Composition	Polypropylene/Polyethylene Blend

**Table 5 materials-16-03903-t005:** Proportions of simple and fiber-reinforced concrete mixes.

Mix	Admixture kg/m^3^			Fiber (kg/m^3^)	PC kg/m^3^	Water L/m^3^	SP %	w/cm	Aggregates kg/m^3^		Slump (mm)
	SF	RHA	TRHA						Gravel	Sand	
Control	0	0	0	0	440	198	0	0.45	930	731	178
F-Control	0	0	0	2.38	440	198	0.05	0.45	930.1	731	178
RHA	0	44	0	0	396	198	0.35	0.45	921	723.5	127
F-RHA	0	44	0	2.38	396	198	0.37	0.45	920.6	723.4	127
TRHA	0	0	44	0	396	198	0.38	0.45	920.1	722.9	102
F-TRHA	0	0	44	2.38	396	198	0.45	0.45	920.6	723.3	114
SF	44	0	0	0	396	198	0.20	0.45	921.8	724.2	102
F-SF	44	0	0	2.38	396	198	0.25	0.45	922.06	724.5	102

PC: Portland cement; HS: silica fume; RHA: heat-treated rice husk ash; TRHA: chemically and thermally treated rice husk ash, F-Control: fiber-reinforced control mixture, F-RHA: fiber-reinforced mixture with heat-treated husk ash, F-TRHA: fiber-reinforced mixture with chemically and thermally treated husk ash, F-SF: fiber-reinforced mixture with silica fume, cm: cementitious material.

**Table 6 materials-16-03903-t006:** Pore structure of cementitious pastes at 56 and 90 days.

Pore Size (mm)	Porosity (%)							
	Control		RHA		TRHA		SF	
	56 Days	90 Days	56 Days	90 Days	56 Days	90 Days	56 Days	90 Days
0.0025–0.005	5.5	7.7	9.5	11.9	11.4	13.4	10.9	13.3
0.005–0.01	7.0	8.5	9.9	12.8	11.2	14.8	11.8	14.2
0.01–0.1	46.1	43.9	49.3	47.9	52.0	49.8	49.9	48.0
0.1–1	31.6	31.2	26.6	23.3	23.1	20.8	24.2	23.2
1–10	5.8	5.6	2.1	2.4	0.9	0.5	1.0	0.7
10–100	3.0	2.0	1.9	1.1	1.2	0.9	1.3	1.0
>100	1	0.8	0.8	0.4	0.22	0.09	0.48	0.23

**Table 7 materials-16-03903-t007:** Mean compressive strengths at different curing days.

Mix	Compressive Strength, MPa				
	7	28	56	90	180
CONTROL	31.5	42.2	48.8	51.1	51.5
SF	35.3	47.6	54.7	55.8	56.6
RHA	32.9	43.8	49.2	52.1	52.6
TRHA	38.6	49.2	55.4	56.9	58.0
F-CONTROL	29.9	40.0	46.3	48.9	49.5
F-SF	33.9	46.4	52.0	54.1	55.3
F-RHA	32.6	41.4	47.0	49.4	50.3
F-TRHA	37.8	48.2	54.4	55.1	56.5

**Table 8 materials-16-03903-t008:** Toughness indices for fiber-reinforced mixtures with and without pozzolanic additions.

MIX	In the First Crack		I5	I10	I20	I30	I60	I10/I5	I20/I10	I30/I20	I60/I30
	Flexural Strength, MPa	Deflection, mm									
FSF	5.27	0.0000856	2.51	3.28	4.47	5.52	8.51	1.31	1.36	1.23	1.54
FTRHA	5.69	0.0000820	2.12	2.65	3.45	4.26	6.84	1.25	1.30	1.24	1.60
FCTRL	4.13	0.0000641	3.34	4.48	5.95	7.17	10.88	1.34	1.33	1.21	1.52
F-RHA	4.81	0.0000861	2.44	3.05	4.06	5.00	7.96	1.25	1.33	1.23	1.59

**Table 9 materials-16-03903-t009:** Chloride penetration test results.

Mix	Adjusted Transferred Load (Coulombs)		Resistivity *ρ* × 10^−3^ (ohms·cm)
CTRL	3529	Moderate	6.25
FCTRL	4189	High	4.92
RHA	1413	Low	14.9
F-RHA	1428	Low	15.0
SF	970	Very low	22.0
FSF	1017	Low	20.4
TRHA	960	Very low	22.2
FTRHA	1005	Low	20.8

**Table 10 materials-16-03903-t010:** Depth of carbonation in concrete mixtures.

Mix	Depth of Carbonation (cm)
CONTROL	0.10
FCONTROL	0.35
RHA	0.20
F-RHA	0.30
TRHA	0.20
FTRHA	0.24
SF	0.15
FSF	0.20

## Data Availability

Data are unavailable due to privacy.

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
