# Peer review of "High Performance Concretes with Highly Reactive Rice Husk Ash and Silica Fume"

_materials, 2023, doi:10.3390/ma16113903_

Round 1

Reviewer 1 Report

1.        Table 1, it is suggested to separate this table into two tables.

2.        Some other papers about high performance concrete are suggested for reference:

doi.org/10.3390/ma16062137

doi.org/10.1016/j.proeng.2011.07.335

3.        Figure 1, is it a SEM image? If yes, amplification factor should be given.

4.        2.2. Aggregates, other properties of fine and coarse aggregates, such as the specific gravity, water content, should be given.

5.        Table 3, some mix codes are hard to understand. For example, it is suggested to change “FRHA” to “F-RHA”.

6.        “For comparison purposes, the water/cement ratio was kept constant at 0.45 for all mixes”. Herein, “water/cement ratio” should be changed to “water/cementitious materials ratio”.

7.        Table 3, I can understand that the SP dosage needed to be regulated to achieve acceptable workability of concrete, but the requirement of workability needs to be stated.

8.        Table 3, “w/(cm)” should be “w/cm”.

9.        Figure 7, it is hard to compare the results of different groups. It is recommended to use colors that are more easily distinguishable for these curves.

10.    3.2.2. Compressive Strength, it is suggested to also add table to present the test results.

11.    Figure 10, it is hard to compare the results of different groups.

12.    Table 6, for the data, some use “.”, some use “,”.

The whole manuscript should be carefully checked for many minor typos.

Reviewer 2 Report

The authors used slica fume and waste rice husk for high performance concrete. The paper is generally good but it needs improvement. Followings should be carried out before acceptance:

The abstract should contain important results of %

It is better that abstract and keyword contain waste/recycled for rice husk

The use of other organic waste should be mentioned into introduction. The following organic waste can be introduced in introduciton: composition component influence on concrete properties with the additive of rubber tree seed shells; normal-weight concrete with improved stress–strain characteristics reinforced with dispersed coconut fibers

How this recycled materials for this study is obtained?

Add sieve analysis results in Figure.

Novelty is not clear. Very same studies are already exists. What is the difference?

The reason for selecting design mixture should be added.

Compare your results with existing studies

Lines 47-49. By replacing, even partially, Portland cement, both economic and environmental benefits are achieved by reducing its consumption and CO2 emissions from the cement industry. For these lines; other types of powder can also be used as cementiotous materials such as glass powder and coal bvottom ash. For this purpose the following studies also should be added: influence of replacing cement with waste glass on mechanical properties of concrete; use of recycled coal bottom ash in reinforced concrete beams as replacement for aggregate; concrete containing waste glass as an environmentally friendly aggregate: a review on fresh and mechanical characteristics; mechanical behavior of crushed waste glass as replacement of aggregates;flexural behavior of reinforced concrete beams using waste marble powder towards application of sustainable concrete; production of perlite-based-aerated geopolymer using hydrogen peroxide as eco-friendly material for energy-efficient buildings; geopolymer concrete with high strength, workability and setting time using recycled steel wires and basalt powder

The type of Figs 4-5 and should be changed. It is not clear to understand

Add photos for test setup?

How did you measaure elastic modulus. Add photos

Add photos for utilized materials. There is no photo related to which materials are utilized.

Fig 10 should be enhanced. It is not readible.

It seems that there is no direct realtion with results. In Fig 4. G15 is lower than 

Please add damaged photos damaged photos of samples

Add some summary for conclucision

Add recent studies on this subject to introduction. There are many studies on the introduction for this topic.

Conclusion should be improved. The recommendation consdiering all test should be given for engineers.

Reviewer 3 Report

This manuscript explores the “High Performance Concretes with Highly Reactive Rice Husk  Ash and Silica Fume”. The manuscript is elaborately described and contextualized with the help of previous and present theoretical background. All the references cited are relevant to this area of research. The methods/analytical study are clearly stated. The result and discussion section are clearly presented. The manuscript needs minor revision and require the following modifications before the acceptance.

1. Abstract – Present the results in the abstract. Presently it only contain conclusion/recommendation.

2. Abstract: This research deals with High performance concrete but there is no single word found in the abstract

2. It would be good if the key words were arranged alphabetically. Reduce the number of key words.

3. Introduction: ‘The line, Pozzolanic materials, for example, have a double effect when combined with Port-55 land cement: they exert a chemical action (pozzolanic reaction) in the cement paste and a 56 filling action that causes the closure of pores and voids’ need to be cited with the following work

https://doi.org/10.1016/j.conbuildmat.2021.123209

https://doi.org/10.1016/B978-0-12-821730-6.00031-0

https://doi.org/10.1002/suco.201800355

4.  Introduction: Present the novelty/problem statement at the end of introduction section.

5. Fig.3. Mark the salient features in the SEM images and then discuss

6. Fig 8,9. Provide error bar

6. Present your research recommendations and scope for future research at the end of the conclusion part.

Dear Editor,

This work can be accepted with some minor modifications

Reviewer 4 Report

SUMMARY

The article submitted for review is relevant. It proposes high-performance concretes with highly reactive rice husk ash and silica fume. The scientific problem of the research lies in the fact that due to the complexity and high cost of obtaining pure silica fume for industrial applications, the search for new sources of high-quality non-crystalline silica fume attracts the interest of researchers from all over the world. The authors see a solution to the problem in the fact that highly reactive silica fume can be obtained from rice husks, an agro-industrial by-product that is quite abundant in some countries. They test the applicability of the obtained silica fume in high-performance concretes and cement pastes. The authors obtained a number of important results by conducting an in-depth study and analyzing the results. The article has practical significance and scientific novelty. The reviewer offers to support this research, but before publishing the article in the journal Materials, the comments should be corrected. They are listed below.

COMMENTS

1.        The authors did not work enough on the abstract. There is no clear definition of the problem. The interest of researchers in obtaining high-quality silica fume is the relevance of the study. I would like to see the problem, that is, what are the deficits and why is there currently an urgent need to obtain high-performance concretes with highly reactive rice husk ash and silica fume? It is important to reflect this at the beginning of the abstract.

2.        It is not clear how the authors describe the methodology. It is necessary not to describe in detail all the laboratory operations performed, but to report that laboratory studies were carried out in the form of experiments, as well as analytical studies in the form of processing the results.

3.        Scientific results are not very clearly expressed. They must be quantifiable. The authors say that the use of TRHA reduces the size and distribution of pores, but does not state by how much. It is reported that concrete mixtures improve their mechanical, strength properties, but it is not reported by how many percent. All this must be added to the abstract.

4.        The authors provide a literature review in the "Introduction" section, but it is not detailed enough. Currently, there are many studies on concretes using silica fume or rice husk concretes. Authors are encouraged to read more literature on the problem under study and supplement the literature review with 5-10 more references. In particular, on the MDPI platform there are a lot of interesting studies about concretes on the ashes of rice husks and rice straw. The "Introduction" section should end with a clear statement of the purpose and tasks of the study.

5.        In the "Materials and Methods" section, figure 1 seems interesting, but it is not described sufficiently in the text. I would like to see a more detailed description of this figure, perhaps with additional signs or symbols applied to it with a callout of explanations outside the figure. What should the reader see in this picture? It is very important.

6.        XRD analysis in Figure 2 needs to be brought to a higher quality, some characters on it are unreadable.

7.        The SEM analysis shown in Figure 3 also contains unreadable characters. I would like to see it in higher quality.

8.        The graph in Figure 7 seems interesting. I would like to see a more detailed explanation of it and also bring it in a larger size and higher resolution. There are similar problems with figure 10. Some characters are indistinguishable or unreadable. The curves in Figure 10b are also poorly distinguished.

9.        The authors present their results in great detail, but do not discuss them in sufficient detail. Perhaps the authors should submit a separate Discussion section after the “Results” section and there provide a detailed comparison of the results obtained with the results of other authors. This will reflect the scientific novelty and scientific significance of the article.

10.     Conclusions should be supplemented by a clear statement of the scientific results obtained, what new knowledge was obtained, as well as proposals for the practical application of the results obtained.

11.     In the list of references, the reviewer draws attention to the presence of a large amount of somewhat outdated data that is more than 30 years old. Concrete technology is a rapidly growing industry and more references should be used over the last 5 years. Authors are encouraged to add 5-10 references for the period 2018-2023.

12.     In general, the conclusion on the article is as follows: the study deserves support, it is interesting and performed at a high level. It is necessary to correct the comments that the reviewer cited, and after that the article can be published in the journal Materials.

Round 2

Reviewer 1 Report

After the review of revised manuscript, I concluded that the authors have taken into account the reviewer's comments, and the manuscript can be accepted for publication.

Minor editing of English language required.

Author Response

We thank the reviewer for useful suggestions and comments and for a quick and insightful review. Track changes have been made to the manuscript. The document has undergone extensive English language editing.

Reviewer 2 Report

The paper can be accepted.

Author Response

We thank the reviewers for their useful suggestions and comments and for a quick and insightful review.